# The AntSMB dataset: a comprehensive compilation of surface mass balance field observations over the Antarctic Ice Sheet

Yetang Wang[1], Minghu Ding[2], Carleen H. Reijmer[3], Paul C. J. P. Smeets[3], Shugui Hou[4], Cunde Xiao[5]

[1]College of Geography and Environment, Shandong Normal University, Jinan 250014, China
[2]Institute of Tibetan Plateau and Polar Meteorology, Chinese Academy of Meteorological Sciences, Beijing 100081, China
[3]Institute for Marine and Atmospheric Research Utrecht, Utrecht University, Utrecht, Netherlands
[4]School of Oceanography, Shanghai Jiao Tong University, Shanghai 200240, China
[5]State Key Laboratory of Earth Surface Processes and Resource Ecology, Beijing Normal University, Beijing 100875, China

*Correspondence to*: Yetang Wang(yetangwang@sdnu.edu.cn), and Cunde Xiao (cdxiao@bnu.edu.cn)

**Abstract.** A comprehensive compilation of observed records is needed for accurate quantification of surface mass balance (SMB) over Antarctica, which is a key challenge for calculation of Antarctic contribution to global sea level change. Here, we present the AntSMB dataset: a new quality-controlled dataset of a variety of published field measurements of the Antarctic Ice Sheet SMB by means of stakes, snow pits, ice cores, ultrasonic sounders and ground-penetrating radars. The dataset collects 268913 individual multi-year averaged observations, 687 annual resolved time series from 675 sites extending back the past 1000 years, and 78968 records at daily resolution from 32 sites across the whole ice sheet. These records are derived from ice core, snow pits, stakes/stake farms, ultrasonic sounders and ground-penetrating radar measurements. This is the first ice-sheet-scale compilation of SMB records at different temporal (daily, annual and multi-year) resolutions from multiple types of measurements, which is available at: https://doi.org/10.11888/Glacio.tpdc.271148 (Wang et al., 2021). The database has potentially wide applications such as the investigation of temporal and spatial variability in SMB, model validation, assessment of remote sensing retrievals and data assimilation.

## 1 Introduction

Under the background of rapid global warming, wide international concerns have been arouse on changes in the Antarctic Ice Sheet (AIS) mass balance, which positively contributed 14.0±2.0 mm to global sea level rise over 1979-2017 (Rignot et al., 2019). Antarctic mass balance is dependent on the partitioning between ice discharge into the ocean and net snow accumulation at the surface, i.e., surface mass balance (SMB). Recent negative mass balance of the ice sheet reflects larger ice dynamical loss than mass gain from SMB (e.g., Shepherd et al., 2012; Shepherd et al., 2018). Despite the responsibility of ice discharge for Antarctic mass balance on the decadal or longer time scales, considerable inter-annual variability is largely determined by fluctuations in SMB (Wouters et al., 2013). Because annual net mass input into the entire ice sheet through snowfall is equivalent to about 6 mm global sea level decline (Church et al., 2001), any small fluctuations in the Antarctic SMB can even result in large variability and trends of global sea level.



SMB is defined as the sum of precipitation, surface and drifting snow sublimation, erosion/deposition caused by drifting snow, and surface meltwater run-off. Since the first international polar year 1957/58 (IPY), a number of scientific Antarctic traverses/expeditions have been performed with the goals of SMB measurements by means of stakes, ice cores/snow pits,

ultrasonic sounders, or ground-penetrating radar (GPR). Due to logistical constraints in the harsh environment, gaps in the spatial coverage of SMB measurements are still large, and long-term samplings are also scarce (Favier et al., 2013). As a result, substantial caveats have been encountered when quantifying SMB at the ice sheet scale by using simple interpolation of these observations (Magand et al., 2008; Genthon et al., 2009). Climate models and various atmospheric reanalysis products provide an important choice to assess SMB for large areas. The outputs of regional climate models have been used

to calculate ice sheet SMB in recent decades by a wealth of Antarctic mass change estimate studies (e.g., Rignot et al., 2011; Shepherd et al., 2012; Rignot et al., 2019). However, these simulations depend on ground-based observations to improve their accuracy and resolution. Before application, the model's performance need to be carefully assessed based on in situ observations, as done by some previous studies (Medley et al., 2013; Wang et al., 2015; Van Wessem 2018; Agosta et al., 2019; Wang et al., 2020). To improve ice sheet SMB estimates, field measurements have been used by cross comparison

with remotely sensed data (Arthern et al., 2006), or outputs of the climate models (e.g., Monaghan et al., 2006; Van de Berg et al., 2006; Medley et al., 2019; Wang et al., 2019). Thus, it is still pivotal to compile all available observations from the past to present to better estimate spatial and temporal variability in SMB, and to constraint climate models and remote sensing algorithm.

Vaughan and Russell (1997) performed the pioneering work to compile all multi-year averaged SMB field measurement data over the AIS, and this compilation was detailly introduced by Vaughan et al. (1999). However, according to Magand et al. (2007), this dataset includes a lot of unreliable data, and should be used with caution. To improve this, Favier et al. (2013) updated the database using the new field measurements carried out during 1999-2012, through a quality control proposed by Magand et al. (2007). Recently, several compilations of SMB measurements at annual resolution have been published (e.g.,

Mayewski et al., 2013; Altnau et al., 2015; Thomas et al., 2017, Montgomery et al., 2018). In spite of numerous field measurements in these datasets, most cover only limited area of the AIS. In particular, these datasets missed a large amount of annually resolved stake/stake farm observations, such as data from the Japanese Antarctic Research Expedition (JARE), South Pole and Vostok, and so on. In addition, available SMB measurements derived from GPR are not or at least not fully collected into these datasets. Furthermore, all available ultrasonic sounder data from automatic weather stations (AWSs) at

daily or higher resolution have not been compiled until now.

In this study, our objective is to generate a comprehensive SMB database for Antarctica, using all available measurements by means of stake or stake network, snow pit or ice core, GPR, and ultrasonic sounder, with the control of data quality. This dataset includes SMB measurements at daily, annual, and multi-year resolutions, which can be applied for validation and



calibration of climate models and remote sensing, developments of remotely sensed algorithm, examination of spatial and temporal patterns in Antarctic SMB and estimate of the drivers of SMB changes across multiple scales. As a case of model validation, we make a comparison of the dataset with ERA5 reanalysis.

## 2 Description of the AntSMB dataset

### 2.1 Data collections and sources

We compile the dataset of SMB measurements over the AIS by searching the literature and public data portal platforms (e.g., the National Snow and Ice Data Center, NSIDC, PANGAEA and World Data Service for Paleoclimatology, NOAA), by collecting the supplements of publications, and by asking individual data generators to contribute their field measurements by email. If two or more request emails were not replied, we consider the data to be unavailable for the public, and thus they are not included in this dataset.


The new data resources of the records in the database include 175373 individual GPR measurements over West Antarctic coastal zones during 2010-2017 (Dattler et al., 2019), 6818 over the Thwaites Glacier in 2009 (Medely et al., 2013), and 1038 along between Dome C and Vostok in 2012 (Le Meur et al., 2018), respectively (Fig.1). A large amount of new stake measurements were acquired by revisiting the traverses from Zhongshan Station to Dome Argus (Ding et al., 2015), from

Syowa Station to Dome Fuji (Motoyama et al., 2015), and between Progress Station and Vostok Station (Khodzher et al., 2014). In addition to a new long-term ice core SMB records at the South Pole (Winski et al., 2019), this dataset includes previously published but unreleased time series of SMB records from ice cores drilled over the Lambert Glacier Basin (Xiao et al., 2001; Li et al., 2009; Ding et al., 2017). Furthermore, an important update of annual resolved SMB data results from the continuous stake network measurements performed at the South Pole, Vostok, and six sites of the transverse between

Syowa Station and Dome Fuji. In addition, this is the first public release of the published high-resolution ultrasonic sounder observations on Berkner Island (Reijmer et al., 1999; Reijmer and Van den Broeke, 2003), Dronning Maud Land (Van den Broeke et al., 2004), East Antarctic Plateau (Reijmer and Van den Broeke, 2003), and Chinese transvers from Zhongshan Station to Dome Argus (Liu et al., 2019), which are very useful for the investigation of intra-annual and seasonal cycles of SMB. The remainder of the database are obtained from existing SMB data compilations, including the multi-year averaged

SMB measurements by Favier et al. (2013) and Wang et al. (2016), time series of ice core records at annual resolution by Mayewski et al. (2013), Altnau et al. (2015) and Thomas et al. (2017), and SMB component measurements over the Antarctic Ice Sheet and Greenland Ice Sheet (SUMup dataset) by Montgomery et al. (2018)..



### 2.2 Selection criteria

In order to establish a comprehensive, complete and quality-controlled AIS SMB product for a variety of scientific
application, quantitative criteria are designed for record inclusion in the database to center on the high-resolution and well-
dated records, and to optimize data spatial coverage. The criteria are as follows.

Firstly, the records must be published through peer-review or public available. The duration and temporal resolution of the
records vary by the measurement types. We select the ultrasonic sounder records with the minimum duration of one year.
For annually resolved archives (ice core, stake and stake network measurements), the duration of records included in this
dataset should be at least 10 years, but smaller than 1000 years. For the multi-year averaged observations, the included
records for average span more than 3 years, which are the minimum number of years for an accurate estimation of the mean
local SMB with the uncertainty of < 10% (Magand et al., 2007).

Secondly, the essential parameters for each SMB data are provided, including location, measurement methodology, data time
coverage, and references to the primary data sources.

Thirdly, to ensure the multi-year averaged SMB data reliable at each site, we select the data determined by the anthropogenic
radionuclides and volcanic horizons with errors of <10%, or stake measurements for more than three years, as suggested by
Magand et al. (2007). The observations based on both stable isotopes and chemical markers, and natural radionuclide are
reliable (Magand et al., 2007), and thus included in the dataset. The available GPR-based snow accumulation rate data are
included, because their uncertainties are <5% at a firn depth of 10 m, and decrease with the increase of the depth (Spikes et
al., 2004; Eisen et al., 2008). SMB records of annual resolved ice cores should be either cross-dated or layer-counted. Their
chronology should include at least two age control points, with one near the youngest part and another near the oldest part of
the time series. Also, they must be confirmed by the data generator. Furthermore, ice core SMB records are corrected for the
impact of firn density and the vertical strain rate profile. The preliminary quality control for AWS snow accumulation data
has been performed by data owner by means of removing the null measurements and physically anomalous snow
accumulation data (i.e., data outside of the initial and final accumulation values). Some high-frequency noises still occur in
the AWS snow accumulation data. To reduce the noises, we discard the data points outside of one standard deviation of a
running daily value as done by Fountain et al. (2010), and Cohen and Dean (2013)..

### 2.3 Types of data measurements collected in the AntSMB dataset

#### 2.3.1 Stakes

Stakes are the easiest and most traditional way to measure SMB. After placing a stake vertically in the snow or ice, relative
variations of snow surface heights over a certain period can be determined by repeated measurements of the distance





between the top of the stake and the surface. Changes in snow heights are multiplied by snow density to yield the corresponding SMB. This simple method has been widely applied over Antarctica by almost all national glacier surveys. However, in most cases, spatial representative of a single stake records is very limited due to small-scale disturbance from post-depositional effects such as the interactions between the stake and local wind. To reduce the related uncertainties, stake lines along a transect or stake farms are often used (e.g., Frezzotti et al., 2005; Kameda et al., 2008; Ding et al., 2011). In

particular, these measurements are useful for the investigation of the spatial distribution of SMB at the scale of < kilometer.

Given the repeated measurements, stake observations are only performed over the easily accessible regions. Due to logistic constraint in the extreme environment of Antarctica, the time span for the measurements usually ranges from 1 year to several years or even more.

### 2.3.2 Snow pits/ice cores

Snow pits and ice cores are used to construct SMB changes in time by determining the age and density of different layers. The dating is dependent on the different time markers preserved in the column of snow pit and ice core. Annual layer is dated through counting of seasonal changes in various parameters including the visual stratigraphy, the oxygen and hydrogen isotopic composition, major chemical ion content, hydrogen peroxide, electric conductivity, and so on. When intergraded

with the prominent horizons of known age from volcanic or radioactive markers, accuracy of dating is largely improved and results in time series of annual snow accumulation. Furthermore, the valuable reference horizons can be also used for the estimation of the SMB between horizons.

In Antarctica, counting annual layer based on the seasonal variations of multiparameter records combining with reference

horizons can calculate annual SMB on the high accumulation zones. However, seasonal cycles can hardly be identified at regions with low accumulation of <100 kg m-2 yr-1, especially for the East Antarctic Plateau. Thus, reference horizon may be the most reliable dating method at the low accumulation area, and only yields a mean SMB between two reference horizons.

### 2.3.3 GPR

GPR maps firn stratigraphy along a profile from the surface, and the radar identified firn layer with equal age along the continuous profiles can allow to gain a detailed insight into SMB patterns. To calculate SMB, the isochronous layers must be well dated, which is usually dependent on complementary depth-age of highly resolved ice core records along the radar profile. During the past few decades, grounded GPR has been widely used for the estimation of spatial variation of recent and historical SMB over Antarctica (e.g., Frezzotti et al., 2007; Anschutz et al., 2008; Müllerr et al., 2010). Most recently,

the newly developed airborne radar systems provide the revolutionized SMB measurements over the Antarctic Ice Sheet (Kanagaratnam et al., 2004, 2007). It can robustly resolve the stratigraphy at the shallow (10 m) to intermediate (100 m)



depths and hence to measure annual and multi-year accumulation rates at the width of hundreds of kilometers along aircraft flight tracks. The systems were firstly developed by the Center for Remote Sensing of Ice Sheets, and flown on the National Aeronautics and Space Administration Operation IceBridge (OIB) campaigns (Leuschen, 2010; Rodriguez-Morales et al.,
2013). The AntSMB database contains records of grounded and airborne GPR observations for the 2009-2019 OIB campaigns.

Relative to point measurements such as stakes, snow pits/ice cores, the advantage of GPR observations is to yield a more accurate representation of spatial variations of SMB. Furthermore, the radar images of deep internal horizons allow us to
quantify long-term variability in SMB. The errors of GPR-based SMB observations are associated with the depth and age of the reflector, and extrapolation of density along the radar profile. The resulting uncertainties were estimated to be about 4% of the calculated SMB at a firn depth of 10 m, and about 0.5% at the depth of 60 m (Spikes et al., 2004).

### 2.3.4 AWS

In Antarctica, some AWSs equipped with ultrasonic sensors measure snow surface height changes by detecting the vertical
distance to the surface. Combining with density observation, snow height changes can be converted to SMB. Despite the poor quality occasionally when the blowing snow or fog happens, this method can continuously yield a high (typically hourly) temporal resolution records of SMB (van den Broeke et al., 2004; Gorodetskaya et al., 2013), which can be utilized to identify individual accumulation/ablation events, to quantify seasonal cycle of snow accumulation, and also to calculate the surface energy balance coupled with other AWS observations.

Same as single ice core or stake observation, AWS measurements represent a single location, and spatial representativity is possibly limited. In addition, after collection of raw snow height data, the temperature-dependent speed of sound correction must be performed. The uncertainty of AWS height measurements is estimated to be ±1 cm or 0.4% of the distance to the surface. This means that the measurements are not sufficient to examine the smaller snow accumulation events usually
occurring on the interior of East Antarctic Plateau.

### 2.4 Structure and metadata

The AntSMB dataset includes three subsets which are composed of multi-year averaged SMB observations from stakes, ice cores and GPR measurements, annual resolved SMB measurements by ice cores, stakes and stake networks, and AWS daily
snow height measurements. To facilitate data reuse, subsampling and re-analysis for scientific research efforts, each record in the three sub-datasets include some essential information, i.e., the name of measurement sites, site locations, measurement method, time coverage of the measurements, and citations. Site locations include latitude, longitude and surface elevation. Each location of the measurements is in units of decimal degrees relative to the WGS84 ellipsoid. As listed in the dataset's





metadata, measurement techniques include firn/ice core, snow pits, stake or stake network, ultrasonic sounder and GPR.
Table 1 summarize the essential information for each measurement. Uncertainties of any measurement methods have been
detailly discussed by Eisen et al. (2008).

Among the three subdatsets, the number of the multi-year mean SMB subdataset is largest, including 265334 unique
measurements by radar isochrones, 2276 stake measurements, and 1303 ice core and snow pit observations (Figure 2). The
majority of these observations (~ 69%) are derived from airborne snow radar measurements in the coastal zone of West
Antarctica and Antarctic Peninsula, Ronne Ice Shelf, South Pole, Pine Island and Thwaites glaciers (Medley et al., 2013;
Medley et al., 2014; Dattler et al., 2019). GPR data from two transects in East Antarctica account for 30% of all
measurements in this subdataset. In most cases, SMB values come from original measurements. However, for the Japanese
traverse route from Syowa Station to Dome F, we updated multi-year averaged SMB by combining new stake surface height
measurements during 2007-2013 with the improved snow density data from Wang et al. (2015).

Annual resolved SMB subdataset contains 687 time series of records, of which 79 records come from the compilation of ice
core snow accumulation by Thomas et al. (2017), and 26 from the shallow firn core records in Dronning Maud Land (DML)
collected by Altnau et al. (2015). Continuous stake surface height measurements at sub-annual resolution are available for
the transverse route from Syowa Station to Dome F since the 1970s (Motoyama et al., 2015). We converted the
measurements to SMB for the subdataset by multiplying the height changes by the improved snow density estimation by
Wang et al. (2015).

AWS snow accumulation data are measured by the determination of the variations of the vertical heights between the sensor
and snow using surface ultrasonic height rangers. The measurements cover a wide range of areas, including coastal zone of
East Antarctica (McMorrow et al., 2001) and West Antarctica (van Lipzig et al., 2004b) with high snow accumulation, the
dry East Antarctic Plateau (Reijmer and Broeke, 2003; van den Broeke et al., 2004), Ross Ice Shelf (Cohen and Dean, 2013),
Berkner Island, Lambert Glacier drain, McMurdo Dry Valleys (Doran et al., 2002). To minimize the impact of changes in
the speed of sound, these data have been corrected using the simultaneous air temperature measurements from the AWSs.

**3. Spatial and temporal analysis of the AntSMB dataset**

**3.1 Spatial coverage of SMB records**

The comprehensive observed SMB database collects SMB field data at the daily, annual and multi-year scales from the
whole AIS. Spatial distribution of the records is uneven within Antarctica. AWS snow accumulation measurements were
obtained at 32 sites, of which ten are located at Dronning Maud Land, seven at the Ross Ice Shelf, and four along Chinese
transverse route from Zhongshan Station to Dome A (Figure 1a). Availability of time series from the annual resolution SMB





subdataset is rich for West Antarctica, Dronning Maud Land, Berkner Island and traverse from Syowa Station to Dome F (Figure 1b). However, large parts of Antarctic interior with low snow accumulation remain undocumented, which is easily understood because the seasonal stratigraphy in ice cores is almost unavailable at the regions with the accumulation of < 70 kg m$^{-2}$ yr$^{-1}$ (Frezzotti et al., 2007; Frezzotti et al., 2013). Compared with SMB compilation by Favier et al. (2013), spatial

coverage of the multi-year SMB subdataset has greatly improved, especially for West Antarctica and the Antarctic Peninsula. Despite the improvement of spatial distribution, SMB records are still poor for the region from the Filchner-Ronne ice shelf via the South Pole to Dome C, and for the coastal zone of East Antarctica.

### 3.2 Temporal variability in the SMB records

The records in the comprehensive SMB dataset cover different time spans, ranging from a minimum of 3 years to a
maximum of 1000 years. The covered time periods are closely associated with the measurement method. AWS provides very high-resolution measurements of snow height changes, but the records generally span only a few years (1-18 years). Although a significant advantage of ice cores is to record SMB changes over the long timespans, it is difficult to perform these observations at a high spatial density. Stake farms are the easiest method to observe SMB, but continuous measurements are available between several years and tens of years, largely due to the logistic constraints in the extreme
Antarctic environment. GPR can detect the local SMB from the last tens of years to about 1000 years along continuous profiles of the snowpack. The temporal resolution of GPR measurements is dependent on the age estimates of reflection horizons, and the resulting records in our dataset ranges from decadal to centennial.

For annual resolved SMB subdataset, of 183 time series from ice core and stake network measurements, 47 span the last 200
years (Fig. 3a). The number of time series peaks during the early 2000s when ice cores were retrieved in Dronning Maud Land (Altnau et al., 2015) and West AIS (Mayewski et al., 2013). Prior to 1800, the number of time series decreases greatly, with only ten with the duration beyond the past 500 years, and five beyond the past 1000 years (Fig. 3a). The sharp decline since the mid-2000s results from a lack of coring efforts. Annual resolved stake measurements cover the past 40 years, peaking from the mid-1990s to the early 2000s (Fig. 3b).


For the multi-year averaged SMB subdataset, 83% of the records with the exception of radar measurements cover <20 years, and 43% span <5 years. Figure 3c presents the distribution of years when these records were measured from 1950 onwards. The distribution of the measurements is relatively even, until the 1990s when the number of samples increase. The temporal coverage of radar observations ranges from 25 years to 185 years.

### 4. Inter-comparison of the different types of SMB measurements

The dataset compiles the different types of SMB measurements including ice cores/snow pits, stakes, ultrasonic sensors and GPR approaches. It is critical to investigate if the resulting data have systematic discrepancy due to the distinct measurement



methods. In particular, the measurements by ice core, stake, and ultrasonic sensor are performed at the centimeter scale, whereas GPR samples at the meter scale. Despite the scale difference, near 100-year averaged GPR measurements agree well

with 5-year averaged single stake at the corresponding locations along the transect near Talos Dome, with the differences of around 10% (Frezzotti et al., 2007). Given that no existing observed SMB dataset can be used as an independent reference to the different types of Antarctic SMB observations, the inter-comparison of SMB determined by different methods at the same or near locations are made, as presented in Figure 4. It is clear that despite the different averaged time coverage, they provide a reasonable match with each other, with the largest discrepancy of <20%, which are consistent with the previous

similar inter-comparison (e.g., Vaughan et al., 2004; Frezzotti et al., 2005; Anschütz et al., 2007). In addition, no systematic errors between the different methods are found.

## 5. Comparison with the previous AIS SMB observation datasets

Here, we present an unprecedentedly comprehensive compilation of SMB observations at the daily, annual and multi-year scales. For the compilation of the multi-year averaged SMB ground based observations including stake/stake farm, snow

pit/ice core, and GPR, we apply the same quality control criteria as used in the compilation by Favier et al. (2013) updated by Wang et al. (2016). Compared with the dataset, our compilation greatly improved the data spatial coverage by updating records using more recent published observations along the margins of West Antarctica, across Marine Byrd Land and Antarctic Peninsula, around the South Pole, between Dome C and Vostok, and along the transects of Progress station– Vostok station, Dumont d'Urville–Dome C, and Talos Dome, etc. We also updated SMB records along the transects of

Zhongshan Station-Kunlun Station and Syowa Station-Dome F based on the recent revisiting measurements. In particular, our dataset provides the first comprehensive compilation of grounded and airborne GPR measurements.

In terms of the collections of annual resolution SMB measurements from ice core, the SUMup dataset focused on the limited ice core records at West Antarctica from the US International Trans-Antarctic Scientific Expedition during the early 2000s

(US ITASE, Mayewski et al., 2013) and several cores drilled in 2010/2011 (Medley et al., 2013), and over the DML and Berkner Island from the European Project for Ice Coring in Antarctica (EPICA, Oerter, 2008a-l). The Antarctica 2k database constructed by Thomas et al. (2017) included 80 ice core records spanning at least 30 years, and the shorter and other ground-based measurement records are omitted. However, our Ant-SMB dataset focuses on the collection of annually resolved snow accumulation records from different kinds of measurements covering the whole ice sheet. As a result, this

dataset contains 175 annually resolved ice core snow accumulation records, 8 stake network measurements covering at least 10 years, and 512 time series of continuous stake measurements spanning more than 18 years.

Previous SMB compilations centered on glaciological observations on annual and longer timescales (e.g., Vaughan et al.,1999; Frezzotti et al., 2013; Thomas et al., 2017), which are useful for the examination of trends and large-scale

variability of AIS snow accumulation. Nevertheless, they do not shed insight on SMB changes at much shorter timescales,





such as synoptic scale and accumulation events. AWSs provide high resolution (typically hour) snow accumulation measurements, which is an advantage to quantify seasonal cycle of SMB, and to examine the synoptic sources of individual accumulation events, relative to the other methods such as snow pits, ice cores, and stakes. Snow accumulation data from individual or several AWSs at the different sectors of Antarctica have been published by some previous studies (e.g.,

Reijmer and Van den Broeke, 2003; Thiery et al., 2012; Cohen and Dean, 2013; Thomas et al., 2015). However, these data have been not well compiled until now. Our dataset is the first attempt to collect all AWS snow accumulation measurements in Antarctica.

## 6. Comparison with ERA5

### 6.1 ERA5 output

Reanalysis utilizes a large amount of observations assimilated into a numerical model to generate a spatially and temporally complete state of the atmosphere. Because the main assimilated data are atmospheric and oceanic measurements, reanalysis outputs are not entirely subjective to the density of surface observations, and thus have the potential to provide important information over the regions with few or even no surface observation. Recent studies have revealed that European Centre for Medium Range Weather Forecasts (ECMWF) interim reanalysis (ERA-Interim) is likely to be the best or among the best

reanalysis dataset for the representation of Antarctic precipitation (e.g., Bromwich et al., 2011; Wang et al., 2016).

ERA5 is the fifth generation ECMWF reanalysis product produced by the Integrated Forecasting System (IFS) Cy41r2 operational in 2016 (Hersbach et al., 2020). Compared with ERA-Interim, a major advantage of ERA5 is much higher horizontal vertical resolutions (~ 31km and 137 pressure levels, respectively), and more enhanced outputs (hourly). Furthermore, IFS Cy41r2 includes a more advanced 4DVar assimilation scheme together with an uncertainty estimation, and

much more observations are assimilated. Detailed improvements can be found in Hersbach et al. (2020). This reanalysis dataset has replaced ERA-Interim, of which updates were stopped on August, 2019. Here, our main objective is to determine if the AntSMB dataset is also capable of representing main features of SMB in space and time, compared to ERA5. Despite the recent release of ERA5 data extending back to 1950, we only use the outputs for the 1979-2018 period, due to the spurious shift of reanalysis outputs in 1979 largely caused by changes in the amount of assimilated observations (e.g., Zhang

et al., 2018; Huai et al., 2019; Wang et al., 2020). Although the problem is likely solved by ERA5, a careful assessment of discontinuity in ERA5 time series is still required before application. However, this is beyond the scope of this study.

### 6.2 A subset of data used for the comparison with ERA5

#### 6.2.1 Multi-year averaged SMB observations

Given that the output of climate models center on climate information since 1979 in Antarctica, it is necessary to define a

special dataset for the model comparison. To match with the coverage period of the models, we only retain observations starting from 1950 onwards in the multi-year averaged SMB subdataset. In particular, we discard observations starting for





the 1950-1978 period with the time coverage of no more than 10 years. Because blowing snow processes are not schemed by ERA5, 190 measurements in blue ice regions (SMB values <0) are excluded. Finally, 191816 multi-year averaged observations are left for model-observation comparison.

### 6.2.2 Annual resolved SMB observations

To estimate the temporal performance of ERA5 for snow accumulation, we use the records from annually resolved SMB subdatabase covering at least 10 years starting from 1979. This results in 159 time series of annual resolved SMB. The representativeness of SMB measurements at a simple site for a region is influenced by local noises from the interaction between wind and local snow surface, especially in the regions with accumulation rate of <120 kg m-2 a-1 (e.g., Frezzotti et al., 2005, 2007; Ding et al., 2011). This can be confirmed by that ERA5 simulated individual records highly correlate with each other (r>0.70), but exhibit a variety of relationships with their corresponding ice core SMB time series at 35 cores on the DML plateau, including significantly negative, positive and insignificant correlations (Fig. 5a). Various linear relationships between the simulated and observed time series are also found over the Berkner Island and Ronne Ice Shelf with high density of cores, with r values ranging from -0.35 to 0.67. At the South Pole, ERA5 shows a significant correlation (r=0.68, p<0.05) with stake farm measurements, but fails to do so with the individual ice core records. A main possibility is that SMB derived from stake networks is less noisy by removing small-scale spatial variability based on the average of a lot of stakes together. To reduce local noise and better assess the performance of ERA5, we first average the individual observation records in the same grid cell, and then stack the averaged time series at the same geographic region. If there are ice core records and stake farm observations in the same location, the measurements of stake farm are utilized. Despite only one core at the top of four ice domes where the local noises are minor (Monaghan et al., 2006a), the records are not discarded in the estimate. Following Frezzotti et al. (2007) and (2013), a single ice core site with accumulations of >700 kg m-2 yr-1 allow the determination of annual SMB at ±10% accuracy, which corresponds to the accuracy derived from the instrumental measurement, and hence the corresponding ice core records are retained. If the records from a single ice core are confirmed to be less local noisy by data owners, we also don't omit them. After the composite and filtering, 48 locations or regions with annual resolved SMB are left to compare with ERA5 simulations.

### 6.3 Spatial performance of ERA5 output

A comparison of the density distribution of ERA5 precipitation minus evaporation (P-E) with the filtered multi-year averaged SMB observations reveals that the multi-year averaged dataset is representative of the entire P-E spectrum of the model at the continental scale (Fig.6a). As shown in Fig.6b, the dataset also represents well the samples elevation distribution of SMB in relation to the whole ice sheet, especially between 200 and 1000m elevations where it was not correctly sampled by the SMB observation dataset compiled by Favier (2013).



ERA5 reveals large spatial gradients of snow accumulation over the AIS (Fig. 7a), with values higher than 1000 kg m$^{-2}$ yr$^{-1}$ at the margins, and lower values (<30 kg m$^{-2}$ yr$^{-1}$) on the hinterland of East Antarctic Plateau. There is a very high correlation between ERA5 output and the observed SMB ($R^2$= 0.93, p < 0.01, which is calculated based on the logarithm of SMB values, due to the lognormal SMB distributions). This suggests a good representation of the major spatial patterns as presented by observations, such as coast-to-plateau SMB gradients. No systematic spatial bias is observed on the West AIS, whereas dry biases occur in most sites of inland East Antarctic Plateau, and wet biases in the section of between 30°W and 150°E of the East AIS margins (Fig. 7b). The mean bias accounts for 6.6% of the average of observed SMB, which is slightly higher than regional climate models (MAR and RACMO2.3p2) (Agosta et al., 2019; Van Wessem et al., 2018). It is obvious that ERA5 robustly capture the sharp decrease in SMB with elevation (Fig.7c). Compared with observation in each 200 m elevation bin, ERA5 is slightly wet below 1600 m elevation, whereas dry biases occur in inland Antarctica with the elevations above 3000 m.

### 6.4 Temporal performance of ERA5 output

The correlation coefficients (r) between ERA5 simulations and SMB observations at 48 locations are shown in Fig.5b. Significant and high correlations are observed at two out of five sites over the Antarctic Peninsula, with r values of >0.7 (p<0.05). Over the WAIS, ERA5 simulations are correlated significantly with observations (r>0.45, p<0.05) at 14 out of 18 sites, and correlation coefficients exceed 0.8 at five sites, suggesting relatively good skills of simulated records for capturing the observed variability in accumulation rates. Significant and positive correlations are present over the plateau, and western and eastern coastal areas of the DML. Correlation coefficients show that a large fraction of inter-annual changes (>70%) in SMB observations at Law Dome of the Wilkes Land. The performances are relatively good for South Pole, Vostok, and Talos Dome. However, no significant or negative correlations are observed at the sites of the Lambert Basin, the Princess Elizabeth Land, middle DML coasts and Adélie Land.

To further assess the temporal performance of ERA5, we use the continuous time series of stake measurements along the JARE traverse route from Syowa station to Dome F. These stake measurements are divided into four subgroups, as done for this traverse route by Wang et al. (2015). Stake measurements in each subgroup are stacked, and then compared with the composites of ERA5 simulations at the respective subgroup (Fig.8). ERA5 overestimates the observed SMB at the coastal and katabatic regions, but underestimates those at the inland plateau region. The modeled records match particularly well with observations at the coastal, higher katabatic and inland plateau regions, with higher r2 values of >0.5. Observed SMB at the lower katabatic region is simulated well by the reanalysis dataset.

Overall, ERA5 fits interannual variability in observed SMB acceptly at most sites over the AIS, and this reveals much of atmospheric circulation is represented by this reanalysis product. Nevertheless, its performance is limited at some sites of Lambert Basin, inland West Antarctica, and parts of East Antarctic coasts. These may result from the limited performance of



ERA5 for the storm frequency related to synoptic-scale circulations, and sublimation because of circulation variations. Detailed interpretation of uncertainty of ERA5 is beyond the scope of this study.

## 7. Data availability

The comprehensive SMB observation dataset is available through a Big Earth Data Platform for Three Poles. The dataset can be downloaded from https://doi.org/10.11888/Glacio.tpdc.271148 (Wang et al., 2021). In this repository, the three subdatasets included in the entire dataset are provided in Excel spreadsheet format together with metadata files.

## 8. Discussion and conclusions

The dataset provides an unprecedentedly comprehensive compilation of SMB observations, with better spatial coverage than previous studies. In particular, our compilation greatly improves spatial density of measurements in the 200-1000 m elevations where are not correctly sampled by the dataset from Favier et al. (2013). However, there is a clear need to increase the spatial density of annual resolved SMB measurements over the inland East AIS, and daily SMB observations over West Antarctica, and 90°-170°E sector of East Antarctica.

This dataset can be used to estimate the temporal and spatial changes in the AIS SMB. A temporal homogeneous climatology of SMB for the second half of the 20th century may be obtained by temporal rescaling of the multi-year averaged SMB subdataset against ERA5 outputs. The available syntheses of time series of records from annual resolved SMB subdataset will allow to investigate regional snow accumulation changes during the past several decades or centuries (Kaspari, et al., 2004; Frezzotti et al., 2013; Altnau et al., 2015; Thomas et al., 2017). The combination of annual SMB subdataset with reanalysis products or the outputs of regional climate models can generate gridded datasets to better constraint the temporal and spatial variability AIS SMB at the different scales (Monaghan et al., 2006b; Medley et al., 2019; Wang et al., 2019). The availability of AWS snow height measurements will allow insights into synoptic and seasonal patterns of SMB, which are vital for ice core dating studies.

In the current study, we have made a comparison between observation data and ERA5 output. As a result, in spite of discrepancy in magnitude, ERA5 represents spatial variations of SMB observations well, and captures a large proportion of the inter-annual variability. Similarly, this dataset can be used to evaluate the quality of other atmospheric reanalyses, and regional or global climate models such as JRA-55, MERRA-2, RACMO2.3, MAR and CESM. Moreover, a high spatial density of stake and GPR measurements along several transections from coasts to inland are included in the dataset, which correctly sample the actual distribution of SMB, and thus allow to provide stringent constraints on the models in these specific regions. Annual resolved SMB observations in the database are also likely to be used as an important input of data assimilation for paleoclimate reconstructions (Dalaiden et al., 2020). The dataset is of vital importance for improvement of



remote sensing algorithm for Antarctic snow accumulation/snowfall rate, such as CloudSat 2C-SNOW-PROFILE product (Palerme et al., 2014; Behrangi et al., 2016).

The scientific community are expected to apply this dataset for Antarctic hydrological studies, model-data inter-comparison and remotely sensed algorithm developments. The cryospheric community are also encouraged to further share their SMB observation data to update this dataset in the future.

**Acknowledgements**

Funding this work was the National Natural Science Foundation of China (41971081), the Strategic Priority Research Program of the Chinese Academy of Sciences (XAD19070103), the Project for Outstanding Youth Innovation Team in the Universities of Shandong Province (2019KJH011) and the Outstanding Youth Fund of Shandong Provincial Universities (ZR2016JL030).

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

.





**Table 1 Brief description of metadata fields used in the Antarctic SMB observations database**

| Column | Name of field in database | Description | Format | Unit |
|---|---|---|---|---|
| Site name | Geo_siteName | Name of the site | Number Code | Number Code |
| Dataset ID | DatasetName | Specific identifier assigned to all SMB records from a given site and publication | Number Code | Number Code |
| Latitude | Geo_latidute | Latitude of the site | WGS84 | Decimal degrees (-90° to 90° |
| Longitude | Geo_longitude | Longitude of the site | WGS84 | Decimal degrees (-180° to 180°) |
| Elevation | Geo_elevation | Elevation of the site | Height above the EGM geoid | m above sea level |
| Variable name | SMB | SMB in millimetre of water equivalent per year | Mass loss is defined as negative | $kg\ m^{-2}\ yr^{-1}$ |
| Method | Method | How each measurement was collected | ____ | ____ |
| Starting date | MinYear | Starting date of measurement, i.e., minimum (oldest) year of each SMB record | Number | Year |
| Ending date | MaxYear | Ending date of measurement, i.e., maximum (more recent) year of each SMB record | Number | Year |
| Citation | Citation | Citation for the first publication presenting the SMB record. | ____ | ____ |





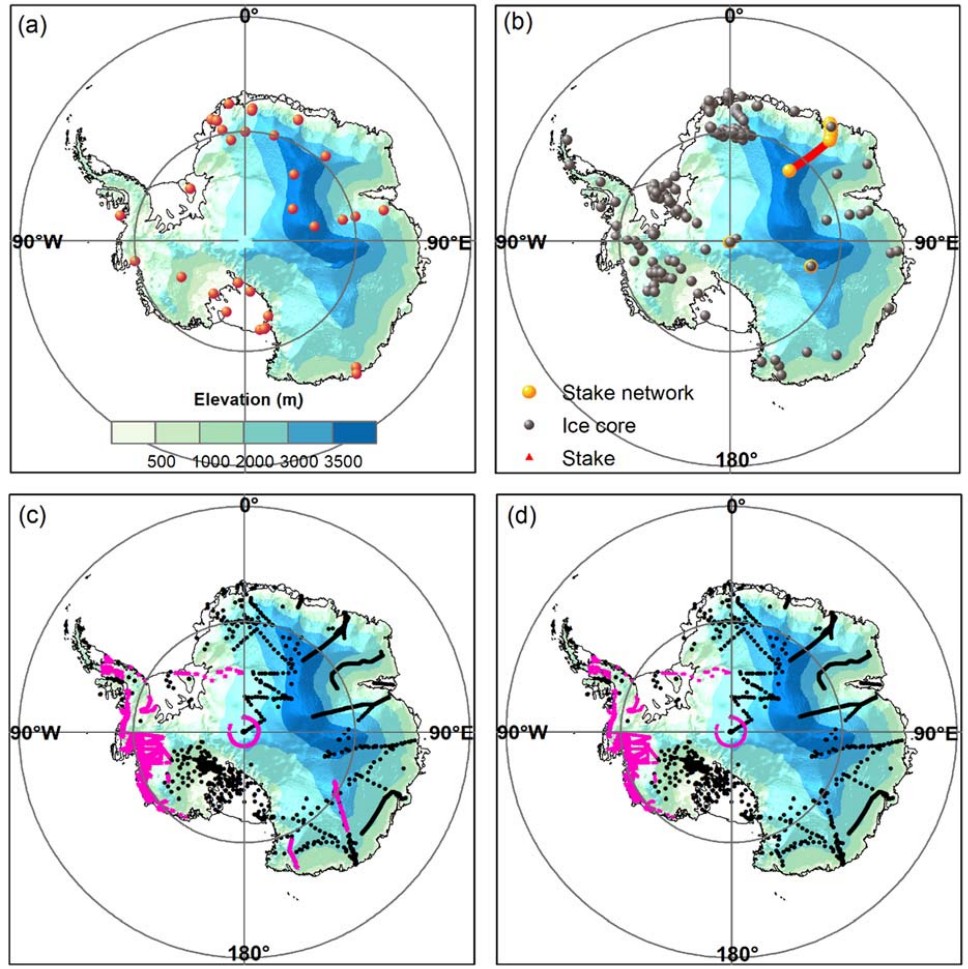

**Figure 1: The comprehensive dataset of Antarctic SMB observations. (a) Spatial distribution of available AWS observations; (b) Locations of available annual resolved SMB observations. (c) Locations of available multi-year averaged SMB field data. Purple five-pointed stars standard for GPR measurements. Black points represent reliable SMB determined by stake/stake farms, ice cores/snow pits. (d) Location of multi-year averaged SMB data only during the second half 20th century selected for model validation.**


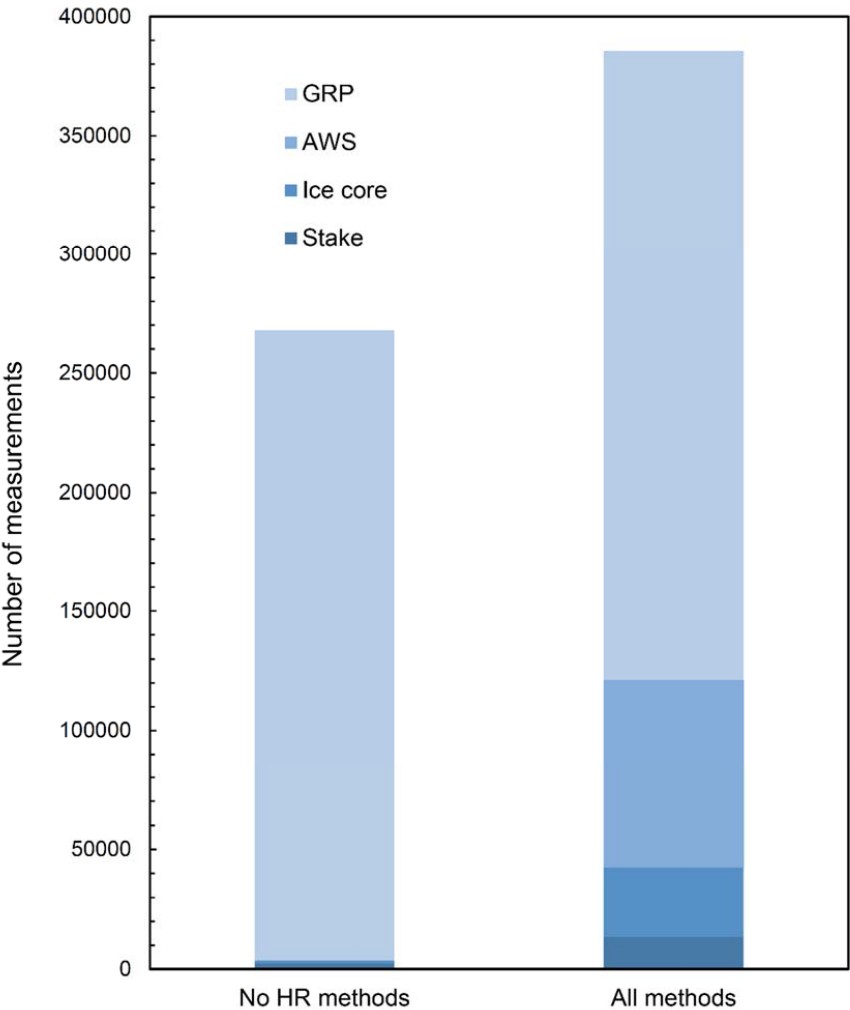

**Figure 2: Bar charts indicating the number of the different types of measurement techniques in the SMB observation dataset. The left bar demonstrates the distribution of approaches with the exclusion of high-resolution snow accumulation measurements, and the distribution of all measurement techniques is shown in the right bar.**


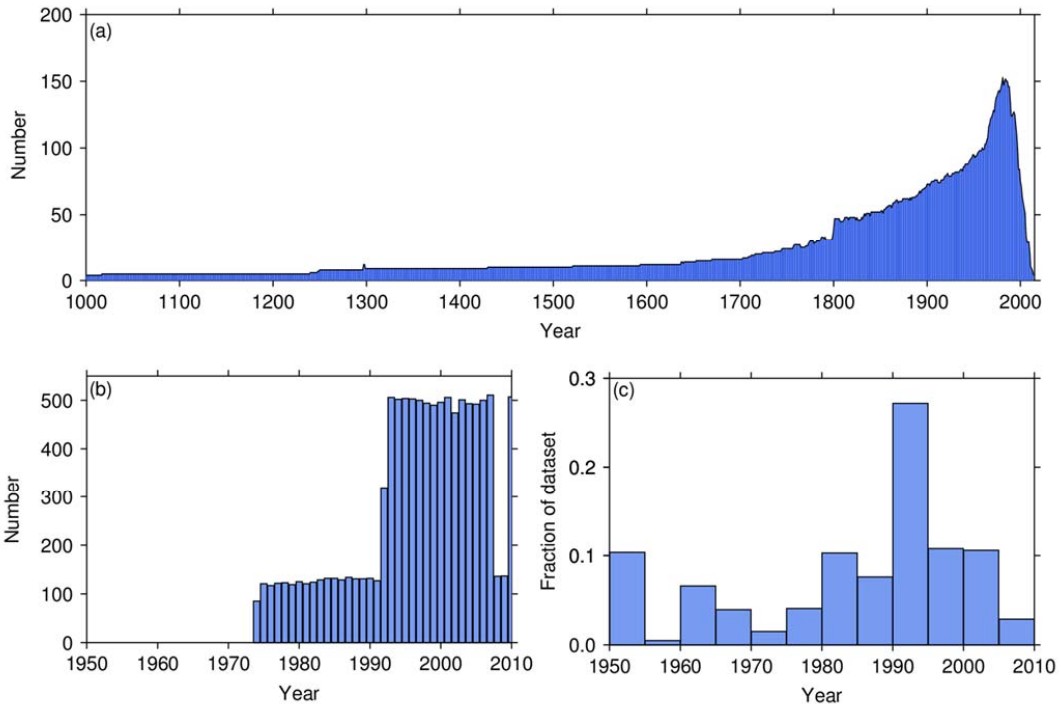

**Figure 3: (a) Availability of records excluding the stake measurements along the traverse from Syowa Station to Dome F in the annual resolved SMB sub-database over time during the past 1000 years. (b)Time coverage of the stake measurements along the traverse from Syowa Station to Dome F. (c) Histograms indicating the date taken of the multi-year averaged SMB subdaset only including ice core and stake measurements.**




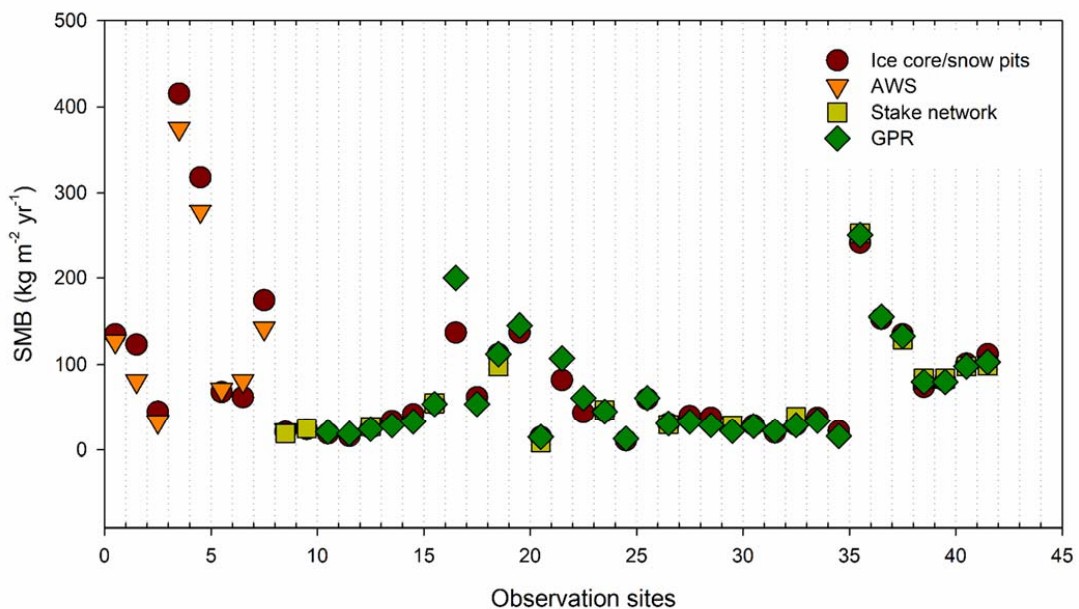

**Figure 4: Inter-comparison between different types of SMB measurements including AWS, snow pit/ice core and GPR at 42 locations.**




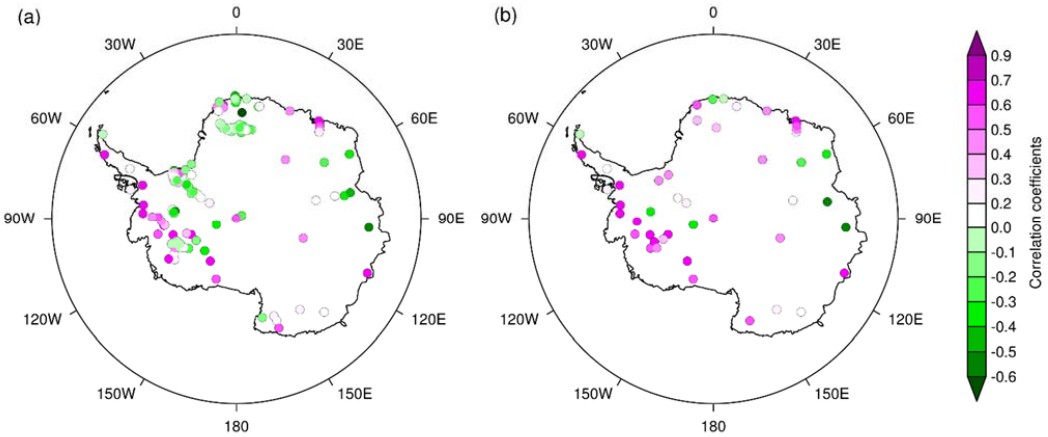


**Figure 5: Spatial distribution of the correlation coefficients (a) between annual resolved SMB observation and ERA5 simulations for their overlapping period, and (b) between averaged observed time series in the same location/region and the corresponding simulations from ERA5.**







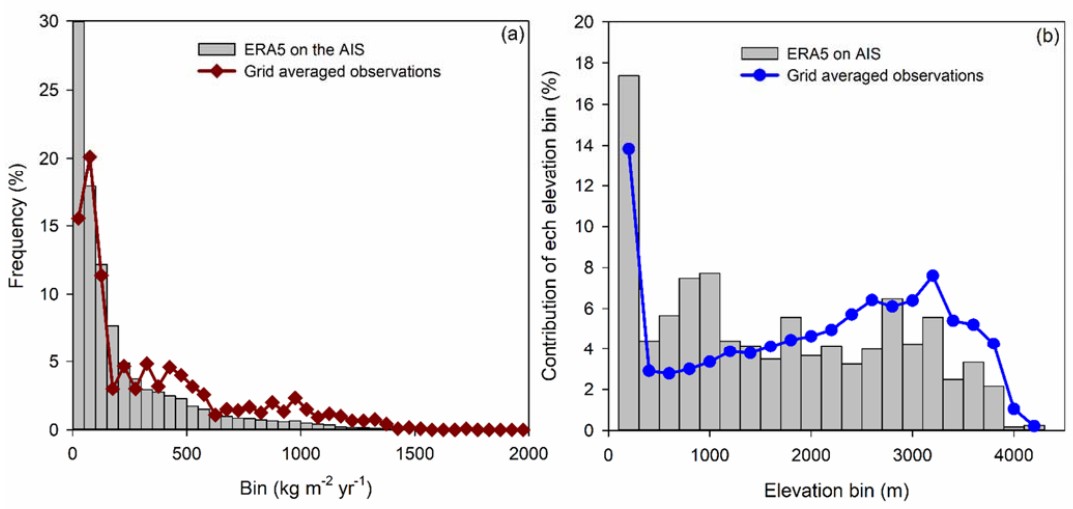

**Figure 6 (a) Relative frequency of ERA5 P-E field over the AIS and gridded averaged records from the multi-year averaged SMB**

**subdatabase, with a bin range of 50kg m⁻² yr⁻¹. (b) The contribution of elevation bin for ERA5 grid cells containing measurements, and entire elevation range to the AIS. The 200m elevation bins are used.**






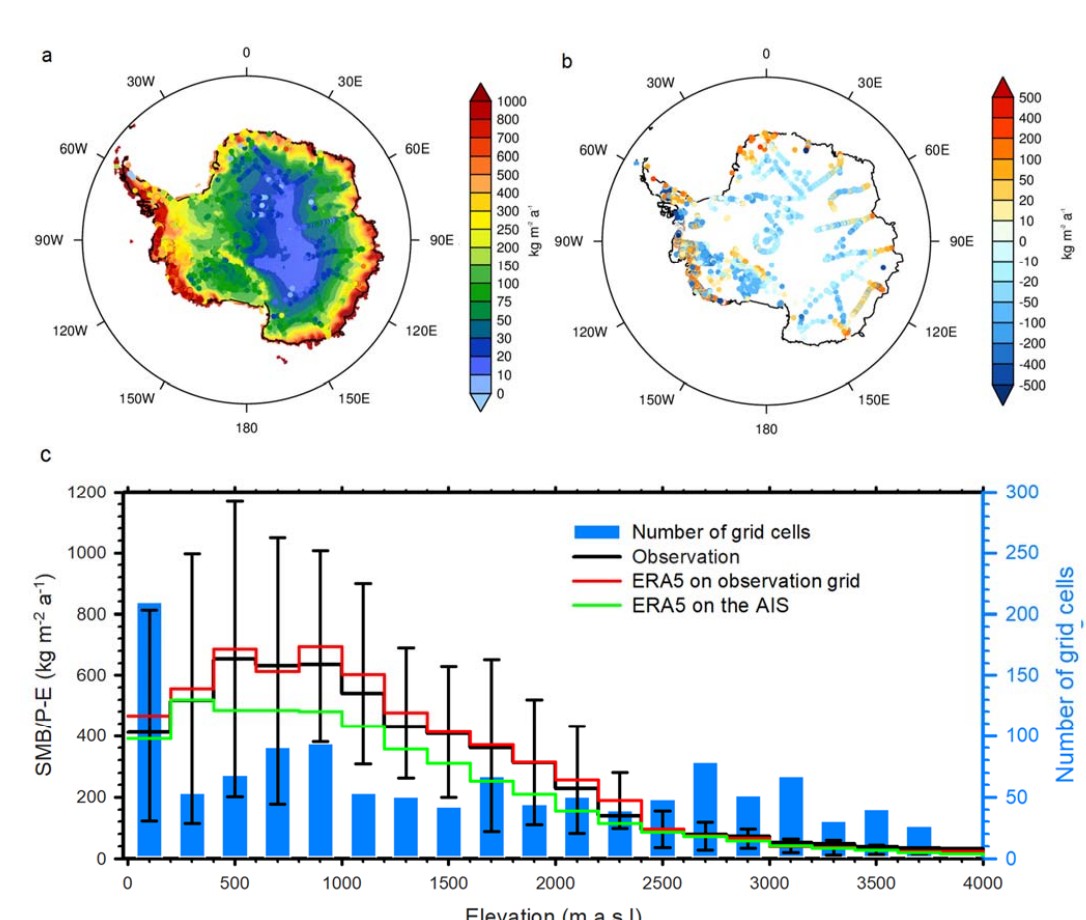

**Figure 7: (a) Spatial distribution of ERA5 mean precipitation minus evaporation (approximated as SMB) for the period 1979–2018, and multi-year averaged SMB measurements. (b) ERA5 minus observed SMB on the ERA5 grid cells, (c) Multi-year averaged observations and ERA5 simulations, binned in 200 m elevation intervals. The number of ERA5 grid cells with in situ measurements in each elevation bin is shown by the blue line (right axis).**


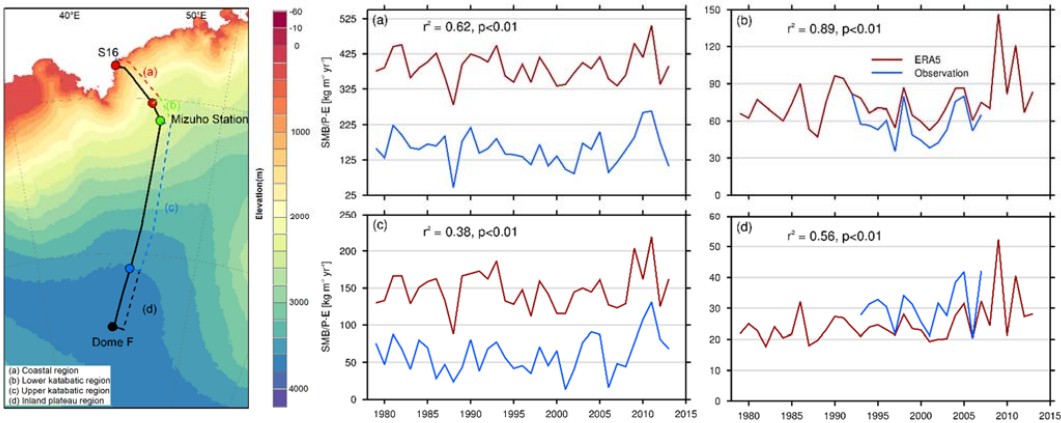


**Figure 8: The left map showing the locations of stake measurements along the traverse between Syowa Station and Dome F, and the regional boundaries. The right four charts showing the comparison of the inter-annual variability in spatially-averaged stake measurements and snow accumulation simulated by ERA5 for (a) the coastal region, (b) lower katabatic region, (c) upper katabatic region, and (d) inland plateau region.**
