# Peer review of "The AntSMB dataset: a comprehensive compilation of surface mass balance field observations over the Antarctic Ice Sheet"

_Earth System Science Data, 2021_

## Author Comment (AC1)

**Response to the reviewer 1**

**Review 1**

This manuscript reported a new compilation of a variety of surface mass balance observations (stake, ice core/snow pit, GPR, ultrasonic sounders), and made it freely available. The compiled data were checked through vigorous quantitative criteria, and thus the relatively reliable data are left. The authors described the features of this new data set, and compare it with previously published observed SMB datasets. They also made a comparison with ERA5 outputs to test the representativeness of the data set for estimating climatic models.

This compilation is a big and complex task, and the resulting data set represent a huge data synthesis effort, which will facilitate a lot of new studies, especially for Antarctic mass balance studies and validation of climate models, and will be well cited. The data set is also interesting for glaciologist, climatologists, geographer, and so on. In my opinion, this manuscript is well written, and deserves to be published. Before publication, only following minor comments needs to be addressed by the authors.

**Response:**

**We are most grateful for the positive evaluation of our work by the referee.**

1. This compilation needs many years of field work by the researchers from different nations, who provide the original data, i.e., the basis of the dataset. Therefore, their scientific contributions are indispensable for this study. However, future work probably simply cite the AntSMB dataset instead of the original studies. So provide the original citation information is important.

**Response:**

**The original citations are included in the three data files of the dataset, which are available at https://doi.org/10.11888/Glacio.tpdc.271148.**

2. The authors mentioned a lot of Antarctic locations in the text, and some might not

easily followed by the readers. So a map with the mentioned locations should be included. Alternatively, this figure is provided in the supplementary material.

**Response:**

**Thanks for your advice. We have added the following figure on the Antarctic locations in the supplementary material.**

[Figure]

**Fig. S1 Map of Antarctica showing the mentioned locations in the text**

3. In section 4, I know that the inter-comparison between different types of SMB measurements in the same locations is helpful. But before making the inter-comparison, please clarify how or based on what did you select these locations? And where are they on Antarctica?

**Response:**

**There are at least two types of SMB measurements at the same or near locations. They are mainly distributed near Talos Dome, along a transect from Terra Nova Bay to Dome C, on the western Dronning Maud Land, and at Dome F and Dome A. Corresponding changes have been made in the text, and the Figure captions.**

**Some other minor issues**

Line 29: please update the reference

**Done**

Line 35: add the reference

**Done**

Line 44: add "the" between " improve" and " ice"

**Done**

Line 89 : "The remainder" is English formulation?

**It has been changed as "the other records".**

Line 146: "100 kg m-2 yr-1" should be " 100 kg m-2 yr-1", and please use the same "unit" throughout the text. For example, in Figure 7, "kg m-2 a-1" was used, but use "kg m-2 yr-1" in the text.

**Corrected throughout the text.**

Line135-149: There is no any reference in the review on SMB measurements from snow pits/ice cores. Please add the related references.

**The references have been added.**

Line183: Change "which are composed of " to "i.e.,"

**Done**

Line 206: This sentence is confusing, and please rephrase it.

**This sentence have been changed as "*We converted the measurements to SMB for the subdataset by multiplying snow height changes by snow density estimated from Wang et al. (2015).*" Hopefully, it is now readable.**

Line 310-311: Not necessarily, since it is due to the assimilation (or lack thereof) of satellite data, and these are mostly lacking before 1979, and thus please delete the sentence.

**Following your advice, and the sentence has been deleted.**

Line 320-375: please use italic "r" and "p"

**Done**

Line 375: Change "r2" to "r2"

**Done**

---

## Author Comment (AC2)

**Response to Reviewer 2**

The authors made a remarkable work to update the current state-of-the-art Antarctic SMB observational dataset, including SMB observations at different temporal resolution and covering different time period, with quality control. This dataset is unique and crucial for evaluating atmospheric models, in the absence of reliable precipitation observations in Antarctica. It is all the more important than SMB is driving the interannual variability of Antarctic ice sheet mass change and that it's evolution under climate change is still highly uncertain, with potential significant contribution to sea level change during the 21st century. In this context, observational reference needed to improve the representation of SMB in atmospheric models is of much value for the whole earth system community.

Consequently, I strongly recommend this article to be published in ESSD.

I have a few suggestions to improve the article further.

**Major suggestions**

My main concern is about the number of observations given throughout the text. I think GPR numbers are not meaningfull because GPR is continuous and the number of points is arbitrary. However it covers large areas. Can you give more meaningfull numbers, such as the length or area covered? A solution would be to bin the GPR data points on a 1 km x 1 km grid and count the number of km2 covered by the data.

**Response:**

**Thanks for your good advice. We have calculated the covered area of GPR measurements by means of binning GPR data points on a 1km×1km grid.**

I thing you should revise the numbers for the following lines:

- p1 L14: "268913 individual multi-year averaged observations"

- p3 L76-78: "175373 individual GPR measurements"

- p7 L193-194: "including 265334 unique measurements by radar isochrones,"

- p11 L318-319 "Finally, 191816 multi-year averaged observations are left for model-observation comparison."

- p7 L194-198: "The majority of these observations (~ 69%) are derived from

airborne snow radar measurements in the coastal zone of West Antarctica and Antarctic Peninsula, Ronne Ice Shelf, South Pole, Pine Island and Thwaites glaciers (Medley et al., 2013; Medley et al., 2014; Dattler et al., 2019). GPR data from two transects in East Antarctica account for 30% of all measurements in this subdatset." + Figure 2

**Response:**

**According to the above calculation, changes have been made accordingly for the number occurring in these lines. Also, we have re-drawn Figure 2. Following is the current Figure 2.**

[Figure]

**Figure 2: Bar charts indicating the number of the different types of measurement techniques in the SMB observation dataset. The left bar demonstrates the distribution of approaches with the exclusion of high-resolution snow accumulation measurements and GPR measurements, and the covered area of GPR measurements is shown in the right bar.**

I think it's not fair to compare stakes numbers with GPR numbers. It makes more sens to compare the areas covered, e.g. using a 1km x 1km grid or a 10km x 10 km grid (number of pixels covered)

**Response:**

**We agree with you, and in the revised version, the GPR covered areas are calculated using 1km×1km grid. Changes have been made accordingly in the text.**

- p11 L342-344 "A comparison of the density distribution of ERA5 precipitation minus evaporation (P-E) with the filtered multi-year averaged SMB observations reveals that the multi-year averaged dataset is representative of the entire P-E spectrum of the model at the continental scale (Fig.6a)."

Are the histograms for observed multi-year SMB including individual GPR points and other kind of data? As stated above, I think it's not fair to compare the number of GPR points with other kind of data, so the histogram might be biases with GPR datapoint numbers.

I suggest to (1) bin GPR data (2) differentiate GPR data from other data in the histograms.

**Response:**

**In the histograms, all the multi-year averaged SMB observations during the 20th century are used. We firstly calculate the averaged SMB for each 30×30 km grid cell (values from points located in the same grid cell are averaged). ERA5 fields are also bilinearly interpolated over a 30 km Cartesian grid. The 30 km grid is chosen due to the spatial resolution of ERA5 (~31km). Then we compare the contribution (relative frequency) of each bin (50 kgm$^{-2}$ yr$^{-1}$ wide) to total SMB. Lastly, we estimate the representativeness of elevation area distribution of SMB observations by the comparison between the contribution of areas with measurements in each 200 elevation bin, and those from ERA5 SMB field. In a word, here we bin all multi-year SMB observations using a 30×30 km grid, and thus, we do not compare the GPR number with the other kind of data.**

Looking at Fig.1c), one can also see that low-elevation datapoints are from GPR and are concentrated on WAIS. I suggest you separate Fig 6 into WAIS and EAIS to discuss the difference of coverage between the 2 basins, and change the conclusion accordingly.

**Response:**

**Thanks for your good advice. Following your advice, we discuss if the AntSMB dataset is representative of P-E spectrum of the model over the WAIS and EAIS, respectively. Following is the re-drawn Fig. 6. Corresponding changes have been made in the text.**

[Figure]

**Figure 6   Relative frequency of ERA5 P-E field data and gridded averaged records from the multi-year averaged SMB subdatabase, with a bin range of 50 kg m-2 yr-1 on (a) the West Antarctic Ice Sheet (WAIS) and (b) the East Antarctic Ice Sheet (EAIS). ERA5 field data are bilinearly interpolated over a 30km Cartesian grid. We average SMB for each 30×30 km grid cell (values from points located in the same grid cell are averaged), and then number of grid cells in each bin are calculated.   The contribution of the area of elevation bin for**

ERA5 grid cells containing measurements, and entire elevation range to (c) the WAIS and (d) the EAIS. The 200m elevation bins are used.

**p11, Section 6.2.2 Annual resolved SMB observations**

→ Can you separate ice cores and other data to assess if there is a real problem specifically with SMB interannual variability in ice cores? It is of importance as annual SMB observation from ice cores have been used to create the widely-used SMB reconstruction from Medley and Thomas (2019).

Response:

As shown in the following Figure 5 of the current version, we separate ice cores from other types of measurements. The corresponding changes have been made in the text.

[Figure]

Figure 5 Spatial distribution of the correlation coefficients (a) between annually resolved SMB observation and ERA5 simulations for their overlapping period (circle: ice core; diamond: stake network) , and (b) between averaged observed time series in the same location/region and the corresponding simulations from

ERA5; (c) standard deviation of observed SMB at annual resolution; (d) standard deviation of annual SMB from EAR5 simulations divided by observations

→ Can you also check for interannual variability (std of annual values) : ice cores vs. other observations vs. ERA5?

**Response:**

**In current version, the comparison of interannual variability in SMB between ice core/other observations with ERA5 for their overlapping period have been added.**

→ You should add a comparison with AWS, at least at some key sites, to assess the usefullness of this dataset.

**Response:**

**Thanks for your good advice.**

**Liu et al. (2019) used AWS snow accumulation observations included in our AntSMB dataset over the Ross Ice Shelf, and inland East Antarctic Plateau to estimate the performance of ERA5 on the synoptic scale. To avoid repetition, we have added a comparison between ERA5 outputs and AWS observations over the Dronning Maud Land. They are as follows.**

*"A recent study showed that ERA5 present relatively good skills for representing snow accumulation changes on the synoptic timescale, observed at the AWSs over the Ross Ice Shelf and along the traverse route from Zhongshan Station to Dome A, with 56%~88% of extreme snowfall events captured (Liu et al., 2019). Given that these AWS observations are included in our AntSMB dataset, to avoid repetition, here we make a comparison between cumulative daily snowfall from ERA5 and the corresponding accumulation records from 11 AWS observations over the DML (Fig. 8). Obviously, gaps in the AWS records occur in most stations because of the problems of sensors or data transmission. Snow accumulation decreases in the daily cumulative AWS records, and reflects the important role of drifting snow,*

*compaction, sublimation or even ablation in the accumulation changes. Despite the noises of these post-deposition processes, stepped increase are observed for both ERA5 snowfall and AWS snow accumulation at each station in Fig. 8. Furthermore, the occurrence of large snowfall events are in broad agreement with the corresponding large accumulation events at all stations. These suggest in spite of the limits of AWS measurements due to the complex impacts of post-deposition noises, they are an important source of ground-based measurements for evaluating synoptic changes in the precipitation from reananlysis products or climate models."*

[Figure]

**Figure 8: Cumulative daily snow accumulation and snowfall over time for each**

station over the Dronning Maud Land (a-k). (l) Spatial distribution of the AWS stations (Notice that AWS3 station records are not included due to a number of missing data)

→ You could add the comparison with ERA-Interim too, to assess the difference of performance compared to ERA5

Response:

It is a good advice to compare with ERA-Interim. However, this manuscript focus on the description of SMB observation dataset, and its potential applications in the future. We take ERA5 as a case to present the application of the database for model assessment. In addition, ESSD emphasize the quality, usability and accessibility of the dataset, not extensive interpretations of data. Thus, the assessment of relative performance of ERA5 and ERA-Interim is outside the scope of this paper, and this journal. In the future, we will carefully estimate the performance of the recent reanalyses such as ERA5, ERA-Interim and MERRA2, and regional climate models such as Polar WRF, RACMO2.3p2 for Antarctic SMB.

**Minor comment and typos**

**Abstract**

- p1 L15: 78968 records at daily resolution from 32 sites across the whole ice sheet

A number of years covered would be more meaningful.

Changes have been made accordingly.

**Introduction**

p1 L28-29: (Wouters et al., 2013) (Church et al., 2001)

Can you update these references with more recent publications?

Done

**2 Description of the AntSMB dataset**

**2.1 Data collections and sources**

- p3 L83: annual

→ annually (everywhere)

**Corrected thoroughly.**

- p3 L85: high-resolution ultrasonic sounder observations

Which density was used to convert to accumulation?

**At the 10 min or 20 min sampling rate.**

**2.2 Selection criteria**

- p4 L97: public

→ publicly

**Corrected**

- p4 L110: "on both stable isotopes and chemical markers, and natural radionuclide"

Can you rephrase? You mean datation based on both stable isotopes and chemical marker?

**This sentence has been rephrased as "*The records with dating based on both stable isotopes and chemical markers, and natural radionuclide are reliable (Magand et al., 2007).*"**

- p4 L111-113 : ". The available GPR-based snow accumulation rate data are included, because their uncertainties are <5% at a firn depth of 10 m, and decrease with the increase of the depth (Spikes et al., 2004; Eisen et al., 2008)."

Does it depend of the post-processing quality? Interpretation of reflectors, proper calibration with ice cores, correct density estimates?

**Response:**

**Yes, this sentence has been changed as "*We also include the available GPR-based snow accumulation rate data, because their uncertainties can be below 5% at a firn depth of 10 m, and decrease with the increase of the depth after post-processing including interpretation of reflectors, correct density estimates, and proper calibration with ice cores (Spikes et al., 2004; Eisen et al., 2008).*"**

- p4 L113-116 "SMB records of annual resolved ice cores should be either

cross-dated or layer-counted. Their chronology should include at least two age control points, with one near the youngest part and another near the oldest part of the time series. Also, they must be confirmed by the data generator. Furthermore, ice core SMB records are corrected for the impact of firn density and the vertical strain rate profile"

Add reference.

**Response:**

**Two references have been added.**

- p4 L116-118: "The preliminary quality control for AWS snow accumulation data has been performed by data owner by means of removing the null measurements and physically anomalous snow accumulation data (i.e., data outside of the initial and final accumulation values)."

Add references.

**Response:**

**Two references have been added.**

- p4 : Third paragraph

This paragraph is difficult to read. It misses a small sentence stating that it is related to quality check. In addition the transition is abrupt between the different kind of data, maybe you could use bullet points? It is all the more true that the different kind of data are described afterwards in Section 1.3

**Response:**

**We agree with you. The sentence on quality check has been added. Bullet points are used for the different kind of observations. Details can be seen as follows, and in the revised text.**

*"Thirdly, the different kind of records are quality-checked to the highest degree as possible, and then selected into the dataset. 1) To ensure the multi-year averaged SMB data reliable at each site, we select the data determined by the anthropogenic radionuclides and volcanic horizons with errors of smaller than 10%, or stake*

*measurements for more than three years, as suggested by Magand et al. (2007). The records with dating based on both stable isotopes and chemical markers, and natural radionuclide are reliable (Magand et al., 2007), and thus included in the dataset. We also include the available GPR-based snow accumulation rate data, because their uncertainties can be below 5% at a firn depth of 10 m, and decrease with the increase of the depth after post-processing including interpretation of reflectors, correct density estimates, and proper calibration with ice cores (Spikes et al., 2004; Eisen et al., 2008).*

*2) SMB records of annually resolved ice cores should be either cross-dated or layer-counted. Their chronology should include at least two age control points, with one near the youngest part and another near the oldest part of the time series (Stenni et al., 2017). Also, they must be confirmed by the data generator. Furthermore, ice core SMB records are corrected for the impact of firn density and the vertical strain rate profile (Thomas et al., 2017).*

*3) The preliminary quality control for AWS snow accumulation data has been performed by data owner by means of removing the null measurements and physically anomalous snow accumulation data (i.e., data outside of the initial and final accumulation values) (e.g., Braaten et al., 1997; 2000). Some high-frequency noises still occur in the AWS snow accumulation data. To reduce the noises, we discard the data points outside of one standard deviation of a running daily value as done by Fountain et al. (2010), and Cohen and Dean (2013)."*

**2.3 Types of data measurements collected in the AntSMB dataset**

- p5 L127-128 : "spatial representative of a single stake records is very limited due to small-scale disturbance from post-depositional effects such as the interactions between the stake and local wind."

spatial representative → spatial representativity

**Corrected**

Not only this but also because of large natural spatial variability .

**Response:**

**This has been added in the sentence.**

▪ p5 L146 "with low accumulation of <100 kg m-2 yr-1,"

→ with accumulation lower than 100 kg m$^{-2}$ yr$^{-1}$ (everywhere)

**Response:**

**Corrected throughout the text.**

▪ p5 L135-148: 2.3.2 Snow pits/ice cores

Add references.

**Done**

▪ p6 L166-167: "The resulting uncertainties were estimated to be about 4% of the calculated SMB at a firn depth of 10 m, and about 0.5% at the depth of 60 m (Spikes et al., 2004)."

I am surprised by such good results. How much does it depend of the calibration?

**Response:**

**According to Spikes et al. (2004), it depends on the calibrations of layer thinning due to ice advection (0 at surface, 1cm at 60m firn depth), depth calibration (10cm at 2m firn depth, 11cm at 60m firn depth), and the isochronal accuracy of each horizon (1 year for all depths). The sentence has been changed as "*The resulting uncertainties were estimated to be about 4% of the calculated SMB at a firn depth of 10 m, and about 0.5% at the depth of 60 m after the calibration of depth and layer thinning, and robustly dating (the isochronal accuracy of about 1 year)    (Spikes et al., 2004).*"**

**2.4 Structure and metadata**

▪ p6 L183-185: "The AntSMB dataset includes three subsets which are composed of multi-year averaged SMB observations from stakes, ice cores and GPR measurements, annual resolved SMB measurements by ice cores, stakes and stake networks, and AWS daily snow height measurements."

Suggestion for readability: "The AntSMB dataset includes three subsets which are composed of (1) multi-year averaged SMB observations from stakes, ice cores and GPR measurements, (2) annually resolved SMB measurements by ice cores, stakes and stake networks, and (3) AWS daily snow height measurements."

**Response:**

**Thanks for your good advice, and changes have been made accordingly.**

- p7 L190-191 "have been detailly discussed"

→ have been discussed in detail

**Changed**

- p7 L201 "Annual resolved SMB"

→ Annually resolved SMB

**Corrected.**

- p7 L209-214: Paragraph on AWS: add the total number of AWS and the number by region. The last sentence of the paragraph is a repetition of "section 2.3.4 AWS"

**Response:**

**The number has been added, and the last sentence has been deleted.**

**3. Spatial and temporal analysis of the AntSMB dataset**

**3.1 Spatial coverage of SMB records**

- p7 L218-220 "AWS snow accumulation measurements were obtained at 32 sites, of which ten are located at Dronning Maud Land, seven at the Ross Ice Shelf, and four along Chinese transverse route from Zhongshan Station to Dome A"

Move the numbers to previous section (consistently with other datasets)

**Response:**

**Following your advice, changes have been made.**

- p8 L223-224 "with the accumulation of < 70 kg m-2 yr-1 (Frezzotti et al., 2007; Frezzotti et al., 2013)."

Here 70, above it was 100?

**Response:**

**It should be "100", and thus is corrected.**

- p8 L229-231 "ranging from a minimum of 3 years to amaximum of 1000 years. The covered time periods are closely associated with the measurement method. AWS provides very high-resolution measurements of snow height changes, but the records generally span only a few years (1-18 years)."

it's contradictory : minimum of 3 years vs. (1-18 years)

**Response:**

**Sorry, it should be 1 year, and corresponding corrections have been made.**

- p8 L237 "resulting records in our dataset ranges from decadal to centennial"

Can't GPR have annual resolution as in Medley et al. (2014)?

**Response:**

**Yes, the resolution of GPR can reach one year. But here we only compiled the multi-year averaged GPR measurements.**

- p8 L239 "For annual resolved SMB subdataset"

→ annually (everywhere)

**Response:**

**Corrected thoroughly throughout the text.**

- p8 L239-240 "of 183 time series from ice core and stake network measurements, 47 span the last 200 years (Fig. 3a)."

I don't know how to read this information from Fig. 3a ? It that the number of measurements in 1800? It seems there is a jump just in 1800?

**Response:**

**It is in 1801, and changes also have been made in the sentence.**

- Figure 3 : add information on the panels for readability : annually-resolved SMB,

multi-year SMB, etc.

**Done**

- p8 L246 "cover <20 years"

→ cover less than 20 years (everywhere)

**Corrected thoroughly throughout the text.**

- p9 L256-258 "Given that no existing observed SMB dataset can be used as an independent reference to the different types of Antarctic SMB observations, the inter-comparison of SMB determined by different methods at the same or near locations are made, as presented in Figure 4."

Did you compare the data for the overlapping time periods or by comparing multi-year averages over different periods?

**Response:**

**We compare multi-year averages over the different timespans.**

- p9 L260-261 "In addition, no systematic errors between the different methods are found."

Aren't AWS giving generally lower SMB than ice cores/snow pits ?

**Response:**

**Indeed, in general, AWS observed SMB are lower than ice cores/snow pits. And this sentence has been deleted.**

- p9 L280 "continuous stake measurement"

What is a "continuous stake measurement" ?

**"Continuous" has been changed as "annual".**

- p10 L299-300 "(ERA-Interim) is likely to be the best or among the best reanalysis dataset for the representation of Antarctic precipitation (e.g., Bromwich et al., 2011; Wang et al., 2016)."

I am surprise as ERA-Interim is known to be too dry inland over the East Antarctic plateau?

**Response:**

*We agree with you, and the sentence has been changed as "ERA-Interim is likely to be the best or among the best reanalysis dataset for the representation of inter-annual variability in Antarctic precipitation (e.g., Bromwich et al., 2011; Wang et al., 2016)."*

- p10 L303 "(~ 31km and 137 pressure levels, respectively)"

Give the same information for ERA-Interim

**Done**

- p10 L306-307 "Here, our main objective is to determine if the AntSMB dataset is also capable of representing main features of SMB in space and time, compared to ERA5."

It must be the reverse : want to know if ERA5 is able to provide a good SMB compared to the AntSMB observational dataset?

**We agree with you, and the corresponding changes have been made in the sentence.**

- p11 L325-327 "This can be confirmed by that ERA5 simulated individual records highly correlate with each other (r>0.70), but exhibit a variety of relationships with their corresponding ice core SMB time series at 35 cores on the DML plateau, including significantly negative, positive and insignificant correlations (Fig. 5a)."

I don't understand this sentence, could you re-write? On the DML plateau,

**Response:**

**This sentence has been rephrased as "*This can be confirmed by that on the DML plateau, ERA5 simulated individual annual SMB highly correlate with each other (r>0.70), but time series of SMB records from different ice cores are poorly correlated, even from the same drilling site. As a result, the relationships between ice core records and the corresponding ERA5 simulations at the drilling core location are variable, including significantly negative, positive and insignificant correlations (Fig. 5a).*"**

- p11 L328 "over the Berkner Island and Ronne Ice Shelf"

Place all locations on a map, at least in Fig. 1

**Response:**

**We have added a figure (Fig.S1) on all Antarctic locations in the supplementary material.**

- p11 L334-336 "Despite only one core at the top of four ice domes where the local noises are minor (Monaghan et al., 2006a), the records are not discarded in the estimate."

I don't understand this sentence, can you clarify?

**Response:**

**We mean that the locations at the tops of ice domes may reduce the amount of small-scale noises in the ice cores. So this sentence has been rephrased as** *"Because the sites at the top of ice domes likely have minor local noises (Monaghan et al., 2006a), the four time series of ice core records from the ice domes are not discarded in the estimate. "*

- p11 L338-339 "If the records from a single ice core are confirmed to be less local noisy by data owners, we also don't omit them."

What do you mean by "less local noisy by data owners" ?

**Response:**

**Sorry for this mistake, and we have deleted this sentence.**

- p12 L351-352 "This suggests a good representation of the major spatial patterns as presented by observations, such as coast-to-plateau SMB gradients."

The correlation can be equal to one and the slope equal to 0.5; correlation cannot be used to assess the good representation of SMB, all the more than the general trend of observations and model is driven by the topography.

**Response:**

*We agree with you, and this sentence has been rephrased as "The major spatial pattern of ERA5 simulations are in good agreement with the multi-year observations (Fig.7a).".*

▪ Figure 7 (a) Avoid using rainbow colormap: https://betterfigures.org/2015/07/10/a-welcome-development-for-matplotlib/

**Response:**

**We re-drawn this figure not using rainbow color.**

▪ p12 L352-352 "No systematic spatial bias is observed on the West AIS"

I can see a wet bias on the coast and dry bias inland?

And a dry bias over Ross ice shelf too?

**Response:**

**We agree with you, and the sentence has been changed as "*Dry biases occur in most sites of inland Antarctica and the Ross Ice Shelf, and wet biases in the ice sheet margins (Fig. 7b)*".**

▪ p12 L370-371 "we use the continuous time series of stake measurements along the JARE traverse route from Syowa station to Dome F."

Aren't these data included in the annually resolved SMB dataset? Why not include them directly in the annual dataset shown in Fig. 5a? Is there other annually-resolved stake lines in your AntSMB dataset?

**Response:**

**Yes, they are included our annually resolved SMB dataset.**

**Part of stake measurements at some years are missing. In particular, at some sites, stake measurements cover only 10 years, and so small samplings are not sufficient for the calculation of correlation coefficients. But spatial average can resolve this issue.**

**Only stake measurements at annual resolution along JARE traverse route between Syowa station and Dome F are included in our dataset.**

p12-13 L380-381 "These may result from the limited performance of ERA5 for the storm frequency related to synoptic-scale circulations, and sublimation because of circulation variations."

Maybe also because of unresolved processes in ERA5 such as drifting snow?

**This has been added in the sentence.**

- p13 L394-396 "A temporal homogeneous climatology of SMB for the second half of the 20th century may be obtained by temporal rescaling of the multi-year averaged SMB subdataset against ERA5 outputs."

Can you clarify? Are you thinking about the kind of temporal rescaling as done by Medley and Thomas (2019) and Wang et al. (2019), as said latter in the paragraph?

**Response:**

**Yes, the method of Medley and Thomas (2019) and Wang et al. (2019) can be used to rescale the multi-year averaged SMB subdataset.**

- p14 L415 "The scientific community are"

→ The scientific community is

**Corrected**

---

## Author Response (AR2)

Dear editor,

This letter accompanies the revision of manuscript essd-2021-22R1, originally entitled by "The AntSMB dataset: a comprehensive compilation of surface mass balance field observations over the Antarctic Ice Sheet".

We would like to thank you for giving us the opportunity to re-revise our paper. We would also like to thank the two reviewers for their further comments, which are very useful for improving our paper. We have carefully revised the paper to take into account all of the points.

In the following, comments are addressed in the same order as in the reviews. The comments are in black fonts and our responses are in blue fonts. We also include the revisions with track-changes as an additional material for reviewing reference. I hope these responses will be helpful to you and the reviewers for re-evaluating our manuscript.

Best regards,

Yetang Wang

**Response to reviewer 1**

Thanks for your further evaluate our paper, and supporting publication.

**Response to reviewer 2**

I thank the authors for taking all my remarks and suggestions into account and for their work.

I think the paper is now ready for publication, except for a few items that I think would need double check:

[major]

Fig. 5d: I am surprised by this result as in Gorte et al. (2020), ice-core standard deviation of annual SMB was estimated much lower than ERA5 standard deviation. Can you double check? Can you show a map of the model standard deviation bellow the observations on Fig. 5c to verify the pattern?

Response:

We have carefully checked Figure 5. Following your advice, a map of standard deviation of ERA5 simulated time series has been added below the observed SMB standard deviation (ice core/stake farm) on Fig.5c.

In Fig.5d, we present the ratios of ERA5 standard deviation to annual observed SMB. Obviously, if the ratio value is smaller than 1, standard deviation of observations is larger than ERA5. At most of sites (>85%), the standard deviation of annual observed SMB is larger than ERA5 standard deviation. In addition, the maximum value of the ratio of standard deviation from ERA5 to observation does not exceed 1.5. We also have redrawn the whole Figure 5.

2. Fig 7a and 7b: there is a problem of consistency between the 2 figures. E.g. observations around South Pole are lower than the model in Fig. 7a, whereas ERA5-observation is negative around South Pole on Fig. 7b. And the problem exists almost everywhere.

The ERA5-observation pattern of Fig. 7b seems very strange, and the dry-wet inland-margin pattern is not seen in Fig. 7c.

I suspect that the difference in Fig. 7b is computed against a constant value of SMB, or something like that?

**Response:**

**Sorry for the errors in the figure, and thanks for your criticism. In the original Fig.7, we use each multi-year averaged measurements including GPR for the 20th century. As the reviewer previously pointed out, GPR is continuous and the number of points are arbitrary. As a result, the observations and ERA5 modelled values on grid cells are not consistent. The difference of a lot of observations at the same grid cell is computed against a constant value of modelled SMB. In the revised version, we has corrected them as follows, and then we have re-drawn Figure 7.**

**ERA5 field data are bilinearly interpolated over a 30km Cartesian grid. If the values from observation points located in the same grid cell (30×30 km), we average them, and extract the corresponding value of this grid cell. At last, we obtain 1217 model–observation comparisons. They are presented in current Fig.7a and b. To make Fig.7c clearer, we have deleted the error bars of observation in each 200m elevation bin.**

**As Fig.7b and c shows, dry biases occur in inland Antarctica, especially on the regions with elevations above 3000m. In addition, parts of Ross Ice Shelf also show dry biases. Wet biases largely are present over the East Antarctic margins.**

You should correct the associated text when you have the corrected map.

**Response:**

**Changes have been made in the text.**

Also there are white dots on Fig.7b whereas there is no white in the colormap?

(e.g. on the Larsen ice shelves)

**Response:**

**They have been corrected.**

[minor]

1.  Fig. 5a and 5b: use symetric scales around 0 and symetric colormap (here the white is on the positive part)

**Response:**

**Thanks for your good advice, and we have redrawn this figure.**

2.  Fig 5d: use darkblue for values higher than 1, as the white is misleading.

**Done**

3. Fig 7a: use a continuous colormap for SMB (e.g. the colormap of Fig. 5c), not a divergent colormap.

**Response:**

**As you suggested, we have changed this using a continuous colormap.**